# TANDEM: Tracking and Dense Mapping in Real-time using Deep Multi-view Stereo

**Lukas Koestler[1*]   Nan Yang[1,2*,†]   Niclas Zeller[2,3]   Daniel Cremers[1,2]**

*equal contribution        †corresponding author

[1]Technical University of Munich     [2]Artisense
[3]Karlsruhe University of Applied Sciences

**Abstract:** In this paper, we present TANDEM, a real-time monocular tracking and dense mapping framework. For pose estimation, TANDEM performs photometric bundle adjustment based on a sliding window of keyframes. To increase the robustness, we propose a novel tracking front-end that performs dense direct image alignment using depth maps rendered from a global model that is built incrementally from dense depth predictions. To predict the dense depth maps, we propose Cascade View-Aggregation MVSNet (CVA-MVSNet) that utilizes the entire active keyframe window by hierarchically constructing 3D cost volumes with adaptive view aggregation to balance the different stereo baselines between the keyframes. Finally, the predicted depth maps are fused into a consistent global map represented as a truncated signed distance function (TSDF) voxel grid. Our experimental results show that TANDEM outperforms other state-of-the-art traditional and learning-based monocular visual odometry (VO) methods in terms of camera tracking. Moreover, TANDEM shows state-of-the-art real-time 3D reconstruction performance. Webpage: https://go.vision.in.tum.de/tandem

**Keywords:** SLAM, Dense Mapping, Multi-view Stereo, Deep Learning

## 1   Introduction

Real-time dense 3D mapping is one of the major challenges in computer vision and robotics. This problem, known as dense SLAM, includes both estimating the 6DoF pose of a sensor and a dense reconstruction of the surroundings. While there exist numerous well-working and robust RGB-D mapping solutions [1, 2, 3], real-time dense reconstruction from monocular cameras is a significantly more difficult challenge as depth values cannot be simply read out from the sensor and fused. Nevertheless, it is a very important problem, as monocular approaches offer significant advantages over RGB-D-based methods [1] which are usually limited to indoor environment due to the near-range sensing or LiDAR-based [4] solutions which are expensive and heavyweight.

Several deep neural network (DNN) based approaches have been proposed to tackle the problem of monocular tracking and dense mapping by utilizing monocular depth estimation [5], variational auto-encoders [6, 7, 8], or end-to-end neural networks [9, 10]. Unlike the aforementioned works, in this paper, we propose a novel monocular dense SLAM method, TANDEM, which, for the first time, integrates learning-based multi-view stereo (MVS) into a traditional optimization-based VO. This novel design of dense SLAM shows state-of-the-art tracking and dense reconstruction accuracy as well as strong generalization capability on challenging real-world datasets with the model trained only on synthetic data. Figure 1 shows the 3D reconstructions delivered by TANDEM on unseen sequences.

**Our contributions. (1)** a novel real-time monocular dense SLAM framework that seamlessly couples classical direct VO and learning-based MVS reconstruction; **(2)** to our knowledge, the first monocular dense tracking front-end that utilizes depth rendered from a global TSDF model; **(3)** a novel MVS network, CVA-MVSNet, which is able to leverage the entire keyframe window by utilizing view aggregation and multi-stage depth prediction; **(4)** state-of-the-art tracking and reconstruction results on both synthetic and real-world data.

5th Conference on Robot Learning (CoRL 2021), London, UK.

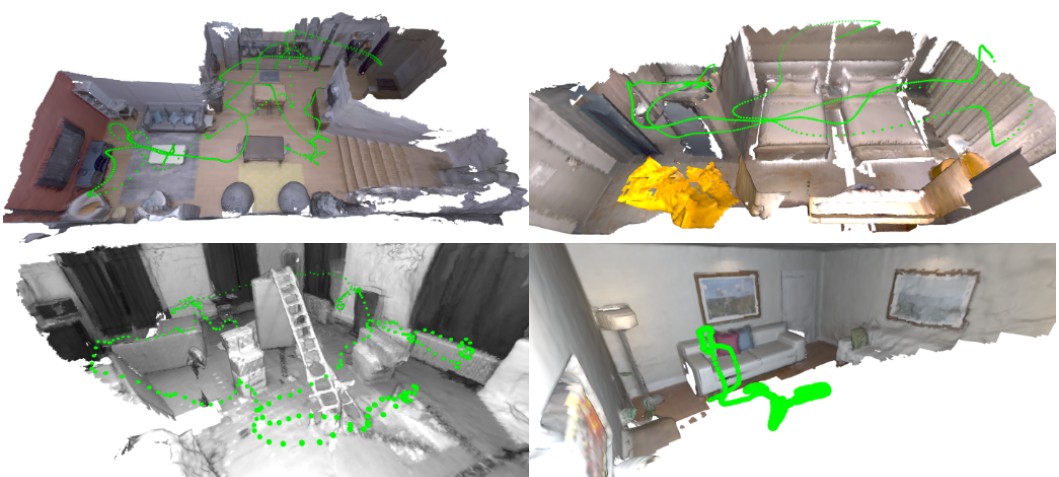

Figure 1: TANDEM is a monocular dense SLAM method that estimates the camera poses and reconstructs the 3D environment in real-time. The figure shows the estimated camera trajectories and the dense 3D models on the sequences of our Replica [11] test split (UL, UR), EuRoC [12] (BL), and ICL-NUIM [13] (BR) using a model trained on the synthetic Replica dataset.

## 2    Related Work

There are two different work streams related to the proposed method. On one side, there is pure 3D reconstruction based on posed images and, on the other side, there are full SLAM or VO frameworks that simultaneously estimate camera poses and a 3D reconstruction of the environment.

**3D Reconstruction.** Most dense 3D reconstruction approaches consider images and corresponding reference poses as inputs and estimate a dense or partially dense reconstruction of the environment. Over the last decade, several classical methods have been proposed [14, 15, 16, 17].

Recently, deep-learning-based methods have shown superior performance over classical methods. These methods regress a 3D model of the environment utilizing DNNs. This 3D model can be either in the form of a volumetric representation [9, 18, 19, 20], a 3D point cloud [21] or a set of depth maps [22, 23, 24]. Nowadays, most popular are methods which predict the final model from 3D cost volumes. Huang et al. [22] proposed one of the first cost-volume-based approaches. Zhou et al. [24] aggregate multiple image-pair-wise volumes to a single cost volume and use a 3D CNN for depth prediction. Yao et al. [23] propose to directly calculate a single volume based on 2D deep feature maps predicted from the input images. In a follow-up work, Yao et al. [25] replace the depth prediction CNN by a recurrent network. To improve run-time and memory consumption, Gu et al. [26] propose a cascade cost volume. Yi et al. [27] introduce a self-adaptive view aggregation to weigh the voxel-wise contribution for each input image. The proposed CVA-MVSNet is built upon the two aforementioned works [26, 27] and largely inspired by them. However, only by their combination and adaption to the SLAM setting we achieve better performance and real-time capability.

While all previous methods are based on a set of selected frames, Murez et al. [9] instead directly predict a TSDF model from a single global 3D cost volume. Weder et al. [18] propose a learning-based alternative to classical TSDF fusion of depth maps. While these volumetric representations, in general, are rather memory intense, Niessner et al. [28] propose voxel-hashing to overcome this limitation and Steinbrücker et al. [29] perform depth map fusion on a CPU using an octree.

**RGB-D SLAM.** In the area of visual SLAM, RGB-D approaches by nature provide dense depth maps along with the camera trajectory and therefore target to solve a similar problem as our approach. Bylow et al. [30] and Kerl et al. [2, 31] mainly focus on accurate trajectory estimation from RGB-D images. In addition to camera tracking, Newcombe et al. [1] integrate the depth maps into a global TSDF representation. Whelan et al. [3] perform surfel-based fusion and non-rigid surface deformation for globally consistent reconstruction. Kähler et al. [32] use a TSDF map representation which is split into sub maps to model loop closures. While most previous methods optimize only for the frame pose, Schöps et al. [33] propose a full bundle adjustment based direct RGB-D SLAM

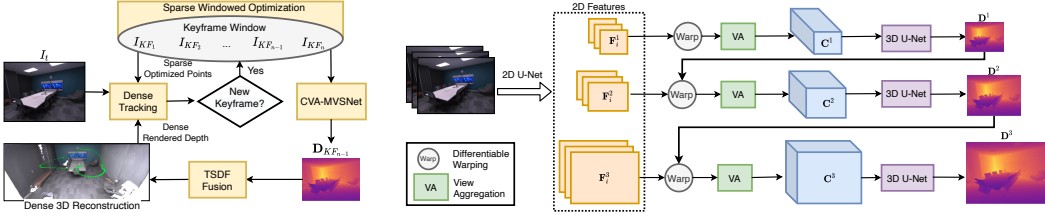

(a) TANDEM System Overview.  (b) CVA-MVSNet Architecture.

Figure 2: (a) Every new frame is tracked using the optimized sparse points from the visual odometry and the dense depth map rendered from the 3D model. The poses of the keyframes are estimated by sliding-window photometric bundle adjustment and fed into the CVA-MVSNet for dense depth prediction. The depth maps are fused into a globally consistent TSDF volume. (b) CVA-MVSNet builds cascaded cost volumes and hierarchically estimates the depth maps. The view aggregation module effectively aggregates the features of the multi-view images by predicting self-adaptive weights.

which optimizes for both camera pose and 3D structure. Sucar et al. [34] integrate a DNN-based implicit scene representation into an RGB-D SLAM system.

**Monocular SLAM.** Compared to RGB-D methods, for monocular approaches both tracking and mapping become much more challenging. Using a single monocular camera, Newcombe et al. [35] perform optimization based on a photometric cost volume to jointly estimate the camera pose and dense depth maps in real-time on a GPU. Engel et al. [36] propose the first large-scale photometric SLAM formulation including loop closure detection and pose graph optimization. By using a sparse representation, Engel et al. [37] were able to formulate the first fully photometric VO framework which jointly estimates pose and depth in real-time. To obtain denser reconstructions, Mur-Artal et al. [38] perform semi-dense probabilistic depth estimation on top of feature-based SLAM [39]. Schöps et al. [40] perform temporal, plane-sweep-based depth estimation using the poses and images obtained from a mobile tracking device. Tateno et al. [5] and Yang et al. [41, 42] leverage DNNs in a traditional direct SLAM framework to improve tracking performance and overcome the problem of scale ambiguity. While traditional geometric and hybrid approaches still achieve superior tracking performance, there are several fully learned SLAM frameworks [43, 44, 45, 24], which are superior in terms of reconstruction completeness. Jatavallabhula et al. [46] propose a differential optimizer which has the potential to bridge the gap between traditional and learning-based SLAM. A novel idea for deep-learning-based SLAM is proposed by Bloesch et al. [6]. The authors propose to learn a frame-wise code representation for the scene depth, which is the jointly optimized together with the camera pose. The work of Czarnowski et al. [7] is an extension of [6], where the code representation is integrated into a full SLAM system. Zuo et al. [8] make use of a similar code representation in a visual-inertial setup and furthermore feed sparse, tracked features into a DNN.

## 3 TANDEM

The proposed TANDEM is comprised of three components: monocular visual odometry (section 3.1), dense depth estimation with CVA-MVSNet (section 3.2), and volumetric mapping (suppl.). Figure 2a shows an overview of the system. The visual odometry utilizes the monocular video stream and the dense depth rendered from the 3D TSDF model to estimate camera poses in a sliding-window manner. Given the keyframes and their estimated poses, the proposed CVA-MVSNet predicts a dense depth map for the reference keyframe. To reconstruct a complete and globally consistent 3D model of the environment, the depth maps are then fused into the TSDF voxel grid [47] with voxel-hashing [28]. By seamlessly integrating these components, the resultant system TANDEM enables real-time tracking and high-quality dense mapping from a monocular camera. Further details, including on the TSDF volume initialization, are given in the supplementary material.

### 3.1 Visual Odometry

Estimating camera poses by tracking a sparse set of points across multiple frames has shown great performance in recent VO systems [37, 39]. Using more points for the joint optimization, however, does not necessarily further improve the accuracy of the estimated poses while significantly

increases the runtime [37]. Therefore, in the proposed VO system we make use of a direct sparse windowed optimization back-end as described in Direct Sparse Odometry (DSO) [37]. However, we utilize dense depth maps rendered from the global TSDF model, which we build up incrementally, in the direct image alignment front-end. In numerous experiments, we confirm that this combination of dense tracking front-end and sparse back-end optimization significantly improves the tracking performance (cf. Table 1) while maintaining a fast runtime.

**Dense Front-end Tracking.** The front-end tracking provides camera-rate pose estimations and serves as initialization for the windowed optimization back-end. In the original DSO, a new frame is tracked against the last keyframe $n$ by direct image alignment using a sparse depth map $\mathbf{D}_n^{\mathrm{DSO}}[\mathbf{p}]$ generated from all points in the optimization window. This approach, however, lacks robustness (cf. Table 1) due to the sparsity of the depth map. We alleviate the issue by incorporating a dense depth map $\mathbf{D}_n^{\mathrm{TSDF}}$ which is rendered from the constructed TSDF model. For each pixel $\mathbf{p}$ in the current keyframe $n$, we assign a depth value either based on the sparse VO points $\mathbf{D}_n^{\mathrm{DSO}}[\mathbf{p}]$, if available, or based on the rendered dense depth $\mathbf{D}_n^{\mathrm{TSDF}}[\mathbf{p}]$, otherwise. Due to the incrementally-built TSDF model, the combined depth buffer might not contain valid depth values for all pixels but it is much denser in comparison to using the sparse depth values only. The nearly-dense combined depth map is used for two-frame direct image alignment.

## 3.2 CVA-MVSNet

Let $\{(I_i, \mathbf{T}_i)\}_{i=1}^n$ be the set of active keyframes where $I_i$ is the image of size $(H, W)$ and $\mathbf{T}_i$ is the corresponding global pose estimated by the VO. CVA-MVSNet is based on the principles of multi-view stereo [48] and further leverages deep neural networks [23] to estimate a depth map for the reference frame $I_{n-1}$. CVA-MVSNet overcomes the prohibitive memory requirement of deep MVS networks by hierarchically estimating the depth using cascaded cost volumes and aggregates the deep features of all the keyframes effectively with a self-adaptive view aggregation module.

As shown in Figure 2b, the multi-scale deep features $\mathbf{F}_i^s$ of the keyframes are firstly extracted by 2D U-Nets with shared weights, where $i \in [1, n]$ is the frame index and $s \in [1, 3]$ is the scale index. As a result, $\mathbf{F}_i^s$ is of the shape $(F^s, H^s, W^s)$ where $F^s$ is the feature dimension of the scale $s$, $H^s = H/2^{3-s}$, and $W^s = W/2^{3-s}$. The depth map of the reference frame is estimated hierarchically with 3 stages each of which takes the set of features $\{\mathbf{F}_i^s\}_{i=1}^n$ as the inputs and predicts the reference depth map $\mathbf{D}^s$ of shape $(H^s, W^s)$. For clarity, we first explain how a single stage estimates the depth and then describe how multiple stages are assembled hierarchically.

**Single Stage Depth Estimation.** For each stage, a cost volume $\mathbf{C}^s$ needs to be constructed using the deep features $\{\mathbf{F}_i^s\}_{i=1}^n$. For each pixel of the reference frame, we define $D^s$ depth hypotheses, which results in a tensor $\mathbf{D}_{hyp}^s$ of shape $(D^s, H^s, W^s)$. The deep features $\mathbf{F}_i^s$ of each frame are geometrically transformed with differentiable warping [49] using the depth hypotheses, the relative pose $\mathbf{T}_j^i = \mathbf{T}_i^{-1} \mathbf{T}_j$ and the camera intrinsics. As a result, a feature volume $\mathbf{V}_i^s$ of shape $(F^s, D^s, H^s, W^s)$ is constructed for every frame.

In order to aggregate the information from multi-view feature volumes into one cost volume $\mathbf{C}^s$, most prior deep MVS methods treat different views equally and use a variance-based cost metric:

$$\mathbf{C}^s = \frac{\sum_{i=1}^n (\mathbf{V}_i^s - \bar{\mathbf{V}}^s)^2}{n}, \quad \text{where} \quad \bar{\mathbf{V}} = \frac{\sum_{i=1}^n \mathbf{V}_i^s}{n}. \tag{1}$$

However, in the sliding-window SLAM setting, the keyframes are not evenly distributed within the optimization window – typically the distance between newer keyframes is much smaller than between older keyframes. This causes considerable occlusion and non-overlapping images. The variance-based cost volume, which weighs different views equally, is thus inappropriate. To alleviate this issue, we employ self-adaptive view aggregation [27] to construct the cost volume:

$$\mathbf{C}^s = \frac{\sum_{i=1, i \neq j}^n (1 + \mathbf{W}_i^s) \odot (\mathbf{V}_i^s - \mathbf{V}_j^s)^2}{n - 1}, \tag{2}$$

where the view aggregation weights $\mathbf{W}_i^s$ have the shape $(1, D^s, H^s, W^s)$ and $\odot$ is element-wise multiplication with broadcasting. The view aggregation weights $\mathbf{W}_i^s$ are estimated by a shallow 3D convolutional network for each $\mathbf{V}_i^s$ separately by taking $(\mathbf{V}_i^s - \mathbf{V}_j^s)^2$ as the input. This aggregation module allows the network to adaptively downweight erroneous information.

The cost volume $\mathbf{C}^s$ is then regularized using a 3D U-Net and finally passed through a softmax non-linearity to obtain a probability volume $\mathbf{P}^s$ of shape $(D^s, H^s, W^s)$. Given the per-pixel depth hypotheses $\mathbf{D}^s_{hyp}$ of shape $(D^s, H^s, W^s)$ the estimated depth is given as the expected value

$$\mathbf{D}^s[h, w] = \sum_{d=1}^{D} \mathbf{P}^s[d, h, w] \cdot \mathbf{D}^s_{hyp}[d, h, w]. \tag{3}$$

**Hierarchical Depth Estimation.** The network leverages the depth estimated from the previous stage $\mathbf{D}^{s-1}(s > 1)$ to define a fine-grained depth hypothesis tensor $\mathbf{D}^s_{hyp}$ with a small $D^s$. Since no prior stage exists for the first stage, each pixel of $\mathbf{D}^1_{hyp}$ has the same depth range defined by $[d_{min}, d_{max}]$ with $D^1 = 48$ depth values. For the later stages $(s > 1)$, the depth $\mathbf{D}^{s-1}$ is upsampled and then used as a prior to define $\mathbf{D}^s_{hyp}$. Specifically, for the pixel location $(h, w)$, $\mathbf{D}^s_{hyp}(\cdot, h, w)$ is defined using the upsampled $\mathbf{D}^{s-1}(h, w)$ as the center and then sampled $D^s$ values around it using a pre-defined offset [26]. In this way, fewer depth planes are needed for the stage with a higher resolution, i.e., $D^1 \geq D^2 \geq D^3$. We train the network using the $L1$ loss applied on all three stages with respect to the ground-truth depth and use the sum as the final loss function.

### 3.3 Implementation Details

To guarantee real-time execution, TANDEM leverages parallelism on multiple levels. Dense tracking and bundle adjustment are executed in parallel threads on the CPU while, asynchronously and in parallel, TSDF fusion and DNN inference are run on the GPU. We train our CVA-MVSNet in PyTorch [50] and perform inference in C++ using TorchScript.

TANDEM can processes images at ca. 20 FPS while running on a desktop with an Nvidia RTX 2080 super with 8 GB VRAM, and an Intel i7-9700K CPU. This includes tracking and dense TSDF mapping but no visualization or mesh generation. We refer to the supplementary for further details, including how to potentially scale the network for deployment on embedded platforms.

## 4 Experimental Results

As TANDEM is a complete dense SLAM system, we evaluate it for both monocular camera tracking and dense 3D reconstruction. Specifically, for the camera tracking, we compare with state-of-the-art traditional sparse monocular odometry, DSO [37] and ORB-SLAM [39], as well as learning-based dense SLAM methods, DeepFactors [7] and CodeVIO [8]. For the 3D reconstruction, we compare with a state-of-the-art deep multi-view stereo method, Cas-MVSNet [26], end-to-end reconstruction method, Atlas [9], learning-based dense SLAM methods, CNN-SLAM [5], DeepFactors [7], and CodeVIO [8], as well as iMAP [34], a recently proposed RGB-D dense SLAM method using a deep implicit map representation [51, 52].

In the following, we will first introduce the datasets we use for training and evaluation. Note that only CVA-MVSNet needs to be trained while the dense tracking part of TANDEM is purely optimization-based and does not require training on specific datasets. Then, the ablation study for TANDEM demonstrates different design choices. In the end, we show quantitative and qualitative comparisons with other state-of-the-art methods. Due to the limited space, we demonstrate only part of the conducted evaluation and refer to the supplementary material for additional experiments.

### 4.1 Datasets

**Training sets**. We train two models for CVA-MVSNet: One on the real-world **ScanNet** [53] dataset and the other one on the synthetic **Replica** dataset [11]. The ScanNet dataset consists of 1513 indoor scenes and we use the official train and test split for training CVA-MVSNet to give a fair comparisons with DeepFactors and Atlas. However, the geometry and texture in ScanNet show noticeable artifacts and incompleteness [11], which limits its potential to training high-quality dense reconstruction methods. To complement the ScanNet dataset, we build upon the recently proposed Replica dataset [11], which consists of 18 photorealistic scenes. These scenes were captured from real-world rooms using a state-of-the-art 3D reconstruction pipeline and show very high-quality texture and geometry. Because the authors did not release any sequences, we extend the dataset by manually creating realistic camera trajectories that yield 55 thousand poses and images.

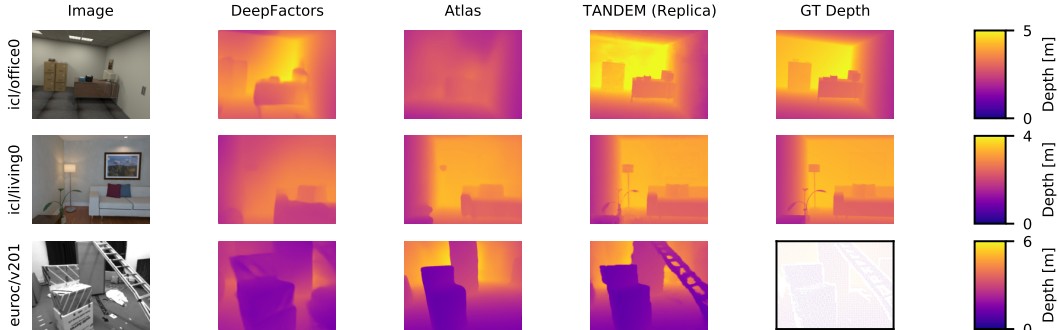

Figure 3: Depth comparison for DeepFactors [7], Atlas [9], and TANDEM on unseen sequences. TANDEM produces finer-scale details, e.g. the plant in the second row, or the ladder in the third row. For EuRoC only sparse ground-truth depth is available. A high-resolution version of this figure can be found in the supplementary material.

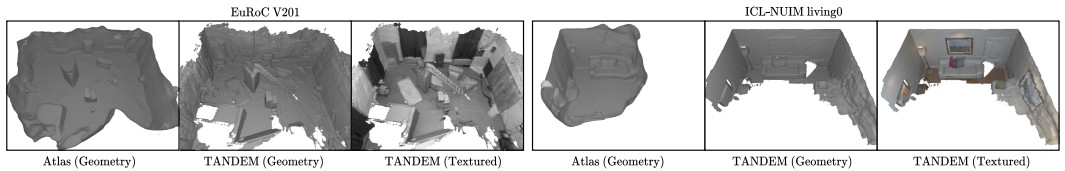

Figure 4: Qualitative comparison of Atlas [9] and TANDEM on unseen sequences. Atlas does not construct textured meshes, so we also render the pure geometry from TANDEM for comparison.

**Evaluation sets**. We use the **ICL-NUIM** [13] dataset and the Vicon Room sequences of the **EuRoC** dataset [12] for evaluating the tracking and the dense 3D reconstruction. Note that TANDEM is not trained on either of the datasets. ICL-NUIM is a synthetic indoor dataset with pose and dense depth ground-truth. It contains low-texture scenes which are challenging for monocular visual odometry and dense depth estimation. EuRoC is a real-world dataset recorded by a micro aerial vehicle (MAV).

## 4.2 Camera Pose Estimation

We evaluate TANDEM for pose estimation against other state-of-the-art monocular SLAM methods on EuRoC and ICL-NUIM. On EuRoC, we evaluate against DSO [37], ORB-SLAM2 [39], Deep-Factors [7], and CodeVIO [8] uses a camera and an IMU sensor. Note that we turn off the global optimization and relocalization of ORB-SLAM2 for a fair comparison. We also implement a variant of DSO (DSO + Dense Depth) which uses all the pixels of the dense depth maps estimated by CVA-MVSNet for the front-end direct image alignment. Note that the difference between TANDEM and DSO + Dense Depth is that TANDEM tracks against the *global 3D model* by rendering the depth maps from the TSDF grid. TANDEM achieves overall better tracking accuracy and robustness than the other monocular methods on both ICL-NUIM and EurRoC. Due to the limited space, we show the results on EuRoC in Table 1 of the main paper and kindly refer to the supplementary material for the results on ICL-NUIM where we also show the comparision with DeepTAM [24].

All the methods except for CodeVIO [8] are run five times for each sequence and reported with the mean RMSE error and standard deviation in terms of absolute pose estimations after Sim(3) alignment with ground-truth. For CodeVIO, we directly take the numbers reported in their paper. The comparison with DSO and DSO + Dense Depth indicates that the proposed dense tracking against the global 3D model improves the camera pose estimations, especially on the more challenging sequences (V102 and V202). However, we should admit that TANDEM still cannot compete with CodeVIO which uses an IMU sensor for pose estimations.

## 4.3 Ablation Study

We conduct the ablation study of CVA-MVSNet on the test split of Replica and show the results in Table 2. Specifically, we evaluate the effectiveness of using the full VO window with 7

Table 1: Pose evaluation on EuRoC [12]. All the methods are Sim(3) aligned w.r.t. the ground-truth trajectories. The mean absolute pose errors and the standard deviations over five runs are shown.

| Sequence | CodeVIO[8] | DeepFactors[7] | DSO [37] | ORB-SLAM2 [39] | DSO+Dense Depth | Ours |
|---|---|---|---|---|---|---|
| EuRoC/V101 | **0.06** (-) | 1.48 ($\pm$0.134) | 0.10 ($\pm$0.006) | 0.31 ($\pm$0.22 ) | 0.09 ($\pm$0.002) | 0.09 ($\pm$0.001) |
| EuRoC/V102 | **0.07** (-) | Lost | 0.27 ($\pm$0.017) | 0.11 ($\pm$0.05 ) | 0.28 ($\pm$0.015) | 0.17 ($\pm$0.006) |
| EuRoC/V201 | **0.10** (-) | 1.06 ($\pm$0.441) | 0.09 ($\pm$0.005) | 1.40 ($\pm$0.211) | 0.09 ($\pm$0.003) | 0.09 ($\pm$0.002) |
| EuRoC/V202 | **0.06** (-) | 1.89 ($\pm$0.019) | 0.21 ($\pm$0.020) | 0.84 ($\pm$0.648) | 0.19 ($\pm$0.022) | 0.12 ($\pm$0.009) |

Table 2: Ablation study of CVA-MVSNet on Replica [11]. Using all keyframes within the VO window (*Win*) does not improve the baseline. However, combining *Win* with view aggregation (*VA*) yields more accurate results at the cost of increased inference runtime and memory. By reducing the number of depth planes (*S*) from $(48, 32, 8)$ to $(48, 4, 4)$ we retain high quality and guarantee the real-time performance of TANDEM. Best shown in **bold** and second best shown underlined.

| | Win | VA | S | Abs$\downarrow$ [cm] | $a_1 \uparrow$ [%] | $a_2 \uparrow$ [%] | $a_3 \uparrow$ [%] | Time$\downarrow$ [ms] | Mem.$\downarrow$ [MiB] |
|---|---|---|---|---|---|---|---|---|---|
| baseline [26] | | | | 2.64 | 98.64 | 82.33 | 20.12 | **142** | 3447 |
| + VO window | ✓ | | | 2.64 | 98.39 | 83.55 | 20.56 | 215 | 4117 |
| + View aggregation | ✓ | ✓ | | **1.92** | **99.00** | **88.59** | 26.03 | 288 | 4117 |
| + Fewer depth planes | ✓ | ✓ | ✓ | 2.33 | 98.51 | 86.93 | **26.09** | 158 | **2917** |

keyframes (Win), the view aggregation module (VA), and fewer depth planes (S) with $(48, 4, 4)$ for $(D^1, D^2, D^3)$. The baseline method is the original Cas-MVSNet using 3 multi-view images as the inputs, no view aggregation module, and more depth planes $(48, 32, 8)$. We use the absolute difference (Abs), and the percentage of inliers with different thresholds ($a_1, a_2, a_3$) as the metrics for the depth map evaluation. Please refer to the supplementary material for the formulas of the metrics. In addition, we also show the inference time and the memory usage of different models. From the table, we can see that using the entire keyframe window with more frames does not improve the accuracy over the baseline model while increasing the runtime and memory usage. With the view aggregation module, the accuracy is significantly improved, but the runtime further increases. Using fewer depth planes does not show a significant drop in accuracy but improves the runtime and memory usage a lot. Therefore, to guarantee the real-time performance of TANDEM, we use the fewer-plane model as the final CVA-MVSNet and all other experiments in the paper are conducted with this model.

### 4.4 3D Reconstruction

We evaluate the reconstruction accuracy on both ICL-NUIM and EuRoC. On ICL-NUIM we compare with DeepFactors [7], CNN-SLAM [5], Atlas [9], and Cas-MVSNet [26]. Table 3 shows the evaluation results. Since Atlas, a pure 3D reconstruction method, does not estimate poses, we provide ground-truth poses as the input. Note that Atlas estimates a TSDF volume directly from a 3D CNN, so we render the depth maps for evaluation against other methods. CNN-SLAM, DeepFactors, and Atlas are trained on ScanNet. For Cas-MVSNet, we re-train it on the Replica dataset and use the same poses as for our CVA-MVSNet. We show the results of the evaluations for our ScanNet-trained model and our Replica-trained model to facilitate a fair comparison. The depth maps of monocular methods are aligned in scale based on the trajectory and further details are given in the supplementary. We use the $a_1$ metric as the major measurement for accuracy. From Table 3, we can see that our method shows a notable improvement in comparison to the other methods and delivers the best result on average. Note that CVA-MVSNet achieves better results with the model trained on the synthetic Replica than the model trained on ScanNet.

On EuRoC, we cannot compare with CNN-SLAM since the numbers on EuRoC are not provided on the paper and code is not publicly available. We further add CodeVIO [8] into the evaluation on EuRoC as it is a recent dense SLAM system and it was also evaluated on EuRoC. Please note that CodeVIO uses a monocular camera and an inertial sensor for tracking, while TANDEM and other SLAM methods rely only on monocular cameras. Table 4 shows the evaluation results.

We further evaluate TANDEM against iMAP [34], an RGB-D dense SLAM system that leverages deep implicit map representation. Please note two major differences between iMAP and TANDEM: on one hand, iMAP uses an RGB-D sensor while TANDEM needs only a monocular camera; on

Table 3: Depth evaluation on ICL-NUIM [13]. We show the percentage of pixels for which the estimated depth falls within 10% of the groundtruth value.

| Sequence | CNN-SLAM [5] | DeepFactors [7] | Atlas [9] | Cas-MVSNet [26] | Ours (ScanNet) | Ours (Replica) |
|---|---|---|---|---|---|---|
| icl/office0 | 19.41 | 30.17 | 28.79 | 77.73 | 52.34 | **84.04** |
| icl/office1 | 29.15 | 20.16 | 62.89 | 88.87 | 61.83 | **91.18** |
| icl/living0 | 12.84 | 20.44 | 83.16 | **97.06** | 86.42 | 97.00 |
| icl/living1 | 13.04 | 20.86 | 36.93 | 84.11 | 71.35 | **90.62** |
| **Average** | 19.77 | 30.17 | 66.93 | 86.94 | 67.99 | **90.71** |

Table 4: Depth evaluation on EuRoC [12]. We show the percentage of correct pixels $d_1$ as in [54].

| Sequence | CodeVIO[8] | DeepFactors[7] | Atlas[9] | Cas-MVSNet[26] | Ours (ScanNet) | Ours (Replica) |
|---|---|---|---|---|---|---|
| EuRoC/V101 | 86.99 | 71.82 | 57.16 | 93.05 | 93.69 | **94.25** |
| EuRoC/V102 | 78.65 | X | **92.29** | 88.03 | 89.62 | 90.50 |
| EuRoC/V201 | 77.32 | 71.85 | 93.28 | 95.43 | 96.84 | **97.17** |
| EuRoC/V202 | 71.98 | 68.26 | 64.33 | 93.16 | 92.65 | **95.68** |
| **Average** | 78.74 | | 76.77 | 92.42 | 93.20 | **94.40** |

Table 5: Comparison to iMAP [34]. TANDEM shows comparable performance to iMAP, which uses RGB-D data, but no training prior to scanning. The mesh-based 3D metrics are as in [34].

| | room-1 | | | office-4 | | |
|---|---|---|---|---|---|---|
| | Acc [cm] | Comp [cm] | CR [%] | Acc [cm] | Comp [cm] | CR [%] |
| iMAP | **3.69** | 4.87 | **83.45** | 4.83 | 6.59 | **77.63** |
| TANDEM | 4.26 | **4.71** | 81.95 | **3.76** | **6.11** | 74.66 |

the other hand, the DNN of iMAP does not require any pre-training and it is trained purely online with the RGB-D inputs, while the depth estimation network of TANDEM requires offline training. Since iMAP was also evaluated on Replica, we, therefore, compare TANDEM with iMAP on the two sequences of their dataset which are not included in the training set of our Replica split. We use the evaluation metrics from iMAP and show the results in Table 5. In general, TANDEM achieves similar results to iMAP while using a monocular camera.

In Figure 3 we show qualitative depth maps estimated by DeepFactors, Atlas, and TANDEM. Both DeepFactors and Atlas can recover the geometry of the underlying scene well, but our method generally manages to capture more fine-scale details. We further show the complete scene reconstruction as meshes in Figure 4. As DeepFactors does not generate a complete 3D model by itself, we only compare TANDEM with Atlas for this experiment. From the figure we can see that, similarly to the depth maps, TANDEM is able to reconstruct more fine-scale details than Atlas.

## 5 Conclusion

We presented TANDEM, a real-time dense monocular SLAM system with a novel design that couples direct photometric visual odometry and deep multi-view stereo. In particular, we propose CVA-MVSNet which leverages the whole keyframe window effectively and predicts high-quality depth maps. Further, the proposed dense tracking scheme bridges camera pose estimation and dense 3D reconstruction by tracking against the global 3D model created with TSDF fusion. The quantitative and qualitative experiments show that TANDEM achieves better results than other state-of-the-art methods for both 3D reconstruction and visual odometry on synthetic and real-world data. We believe that TANDEM further bridges the gap between RGB-D mapping and monocular mapping.

**Acknowledgments**

We thank the anonymous reviewers for providing helpful comments that improved the paper. We express our appreciation to our colleagues, who have supported us, specifically we thank Mariia Gladkova (Technical University of Munich) and Simon Klenk (Technical University of Munich) for proof reading, as well as, Stefan Leutenegger (Technical University of Munich & Imperial College London) and Sotiris Papatheodorou (Imperial College London) for their help with setting up a demo for the conference. We thank the authors of iMAP, specifically Edgar Sucar (Imperial College London), for providing us with evaluation data and scripts from their paper.

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
