# OpenReview forum: "TANDEM: Tracking and Dense Mapping in Real-time using Deep Multi-view Stereo"
_robot-learning.org/CoRL/2021/Conference — CoRL2021 Poster_

### Official Review · Reviewer_W394 · 2021-07-18

**Originality:** Very Good
**Technical Quality:** Very Good
**Clarity Of Presentation:** Very Good
**Impact:** 4

**Recommendation:**

Strong Accept: I recommend accepting the paper and will argue for my recommendation even if other reviewers hold a different opinion.

**Summary:**

This paper presented a new real-time dense SLAM system and achieves SOTA performance in both camera pose estimation and 3D reconstructions. It is built based on a combination of direct sparse odometry (DSO) and deep multi-view stereo network. It shows very good quantitative and qualitative results.

**Issues:**

* Evaluation in the 3D reconstruction: as the authors mentioned in Section 4.1 and 4.4, ScanNet dataset contains more artefacts and holes than Replica dataset. The proposed method was finetuned on the Replica dataset while the compared methods, i.e. DeepFactors, CNN-SLAM, Atlas were all trained on the ScanNet only. It will be a more fair comparison if the paper can also show the performance trained only on the ScanNet in order to remove the effect of the training dataset.
* Evaluation in the camera pose estimation: the front-end tracking component in the compared VO/SLAM methods do not contain much about the learning component as the main innovation in the deepfactor is about its joint optimisation capability. Did the author try comparing with the more recent learning-based VO/SLAM system, such as the [25], and [43] that are mentioned in its related work section?



**Reviewer Expertise:**

Very good: Comprehensive knowledge of the area

**Strengths And Weaknesses:**

The paper is well written in general and easy to follow. It shows impressive results on camera pose estimation and 3D reconstruction and demonstrates good generalization results in unseen environments. The theoretical innovation is based on a combination of existing work. Nevertheless, the authors showed a well-designed system and explained the idea behind it.

It will be better if some minor issues that are mentioned in the issue section could be addressed.

**Summary Of Recommendation:**

The presented paper showed very good results and contains sufficient engineering efforts to make it robust and run in real-time.

---

> ### Author Response · Authors · 2021-08-26
> **Response to Reviewer W394**
>
> We thank the reviewer for their helpful and sharp comments. Based on the reviewer’s recommendation we add a comparison with DeepTAM to the supplementary material and compare with D3VO in our response. We will update the main paper if the paper is accepted such that we can aggregate all suggestions and change the layout of the main paper only once.
>
>
> **Camera Pose Estimation**
>
> We agree with the reviewer that a comparison to DeepTAM [25] and D3VO [43] is of interest and will present our results below. Nevertheless, we would like to highlight that we compare against CodeVIO [8], a visual-inertial odometry system boosted by Deep Neural Networks, that has been recently published in ICRA 2021.
>
> _Camera Pose Estimation (DeepTAM)._
> The official DeepTAM code does not implement the full method with tracking and mapping, but only a tracking-only mode which requires ground truth depth maps for tracking. We consider the re-implementation of the full tracking and mapping mode out of scope for this rebuttal, especially because other users have failed according to an open github issue: https://github.com/lmb-freiburg/deeptam/issues/9.
> For the ICL-NUIM dataset, we compare TANDEM against DeepTAM’s tracker by providing ground truth depth maps to DeepTAM. In Table 1 of the DeepTAM paper, the full tracking and mapping pipeline has approximately twice the error in comparison to tracking with ground truth depth maps. The results are shown in Table 1 in the supplementary material. For the DeepTAM tracker, we used only one run because the result was consistent across multiple runs due to the provided ground truth depth maps. The mean ATE for TANDEM is better on all four ICL-NUIM sequences and for 17 out of the total 20 runs TANDEM achieves a better result than the DeepTAM tracker, while not using ground truth depth maps.
> For the EuRoC dataset, the ground-truth depth maps are sparse and the tracking-only implementation of DeepTAM did not work properly with the sparse depth maps. Therefore a comparison on EuRoC was not possible.
>
> _Camera Pose Estimation (D3VO)._
> D3VO does not have published code, but the paper shows results for the EuRoC dataset. For the sequence V202, D3VO achieves an ATE RMSE of 0.05 m (see D3VO Table 6), while TANDEM achieves an ATE RMSE of 0.12 m. However, D3VO uses the sequences MH01, MH02, MH04, V101, V102, and V201 for training even including sequences from the same room as the testing sequence. On the other hand, TANDEM is trained only on the Replica dataset. Additionally, if D3VO is trained on all machine hall (MH) sequences it achieves a depth accuracy (d_1) of 43.8% (see D3VO Table 3) for V201 while the Replica-trained TANDEM achieves 97.17%. This shows that D3VO is geared towards estimating accurate poses and TANDEM is focused on 3D reconstruction. Finally, the advances made in D3VO like the self-supervised training scheme, the pose network, and the uncertainty network could be integrated into TANDEM to obtain even more accurate pose estimates.
>
>
> **3D Reconstruction Evaluation**
>
> We thank the reviewer for their comment and would like to point out that we compare with a model trained solely on the ScanNet dataset (Table 3, “Ours (ScanNet)”), which achieves the second-best performance after our model trained on the Replica dataset. We will make this more explicit in the paper.

---

> > ### Comment · Reviewer_W394 · 2021-09-02
> > **Thanks for solving the concerns**
> >
> > I would like to thank the authors for solving my concerns. Despite the limited theoretical contributions and analysis in this work, the engineering efforts in making this system generalizing to unseen environments and real-time capability can make a contribution to CoRL.

---

### Official Review · Reviewer_xCmq · 2021-07-23

**Originality:** Very Good
**Technical Quality:** Very Good
**Clarity Of Presentation:** Very Good
**Impact:** 3

**Recommendation:**

Weak Accept: I recommend accepting the paper, but will not argue for my recommendation if the majority of other reviewers have a different opinion.

**Summary:**

This paper proposes a new dense SLAM system. The key novelty lies in the system design - a classical direct VO method is used for the tracking and a learned multi-view stereo (MVS) module is used for the dense mapping. This combination improves both the tracking accuracy (as the VO has access to dense depth maps) and the mapping accuracy as learned MVS is known to be very accurate. The results presented in the paper support this hypothesis.

**Issues:**

I expect the reviewers to address the issues above.

**Reviewer Expertise:**

Very good: Comprehensive knowledge of the area

**Strengths And Weaknesses:**

**Strengths**
- The paper is clearly written and the method is well explained. The figures and the writing are polished and of publication quality.
- The system design is novel in the sense that although multi-view stereo is closely related to dense SLAM, these approaches have largely been developing independently and have not been utilized for realtime SLAM. This is mainly because they used to be really computationally intensive, however, with the advent of deep-learning based multi-view stereo, such as MVS-Net these approaches have been much more efficient and thus utilising them in SLAM is a natural choice.
- The evaluation in the paper is solid. The approach is compared to a number of recent systems allowing the reader to get a sense of its performance.
- The results show that the system is accurate in both tracking and mapping. It achieves SOTA in terms of tracking and mapping accuracy on challenging datasets such as EuRoC.

**Weaknesses**
- There doesn’t seem to be any explanation of how the depth is initialized. This is quite important as the tracking relies on the depth rendered from the TSDF model, but at the start no model would have been built. Initialization is a very important problem in SLAM and I would expect some discussion on this point in the paper.
- I can’t see a comparative evaluation of the runtime performance. There are some times reported for individual components, but what is the overall performance compared to other systems? I assume this is realtime as the approach is compared to other realtime systems in the evaluation. However, I think it is important that the authors include an overall runtime comparison. This is because most SLAM applications need realtime performance.
- There are no examples of failure cases or challenging examples, like rotation-only motion. This is important as the system surely fails in certain situations and it would be good to know what these are.


**Summary Of Recommendation:**

If the weaknesses above can satisfactorily be addressed, I recommend accepting the paper. The main reason is that I think it is a well-designed, clean system that achieves outstanding performance compared to the current state of the art.

---

> ### Author Response · Authors · 2021-08-26
> **Response to Reviewer xCmq**
>
> We thank the reviewer for their helpful comments and address the initialization, the comparative runtime analysis, and the failure cases below. We have added significant new material to the supplementary material and will update the main paper if the paper is accepted such that we can aggregate all suggestions and change the layout only once.
>
>
> **Initialization**
>
> We thank the reviewer for their comment and have added Section 2 “Initialization”, Figure 5, and Figure 6 to the updated supplementary material. The proposed CVA-MVSNet operates on a keyframe window and thus the tracking is initialized the same way as in DSO with non-linear optimization on poses and the sparse depth. The TSDF volume is initialized to represent empty space, i.e. with zero weights W(x) = 0 forall x. Because we use voxel hashing to represent the TSDF volume, empty space can be represented very efficiently by not allocating any voxel blocks. After the first dense depth map is predicted and integrated into the TSDF} volume, TANDEM uses the rendered nearly dense depth maps for tracking as can be seen from Figure 5 in the supplementary material.
>
>
> **Comparative Runtime Analysis**
>
> Overall, the analysis shows that on the same hardware TANDEM and DeepFactors have approximately the same runtime and DSO runs considerably faster.
>
> _Comparison with DSO._
> On the same sequence and system that we used for the TANDEM timing experiment, DSO achieves a throughput of ca. 56 FPS. The higher throughput is expected given that DSO performs no neural network inference and no dense tracking.
>
> _Comparison with DeepFactors._
> We time DeepFactors on the same system as TANDEM using their `--flagfile=data/flags/live_odom.flags` configuration. To the best of our knowledge, their code does not offer any flags to turn off the GUI. We ignored the start-up time for a fair comparison. On their demo scene (scannet/scene0356_02) their system runs with ca. 21 FPS and the output in the console indicates that a small fraction of that time is spent for rendering the GUI. This seems consistent with the claims in their paper. However, the runtime is highly scene dependent and DeepFactors runs with approximately 13 FPS on EuRoC/V101, which is the scene we used for benchmarking TANDEM (ca. 21 FPS). It should be noted that DeepFactors also saves numerous files to disk if it crashes so the corresponding information must be stored in RAM, even if it is not written during a successful run. Overall, the comparative analysis with DeepFactors indicates that both systems run at approximately real-time rates and we refrain from further statements given the intricacies of benchmarking and missing performance flags for DeepFactors.
>
>
> **Failure Cases**
>
> We thank the reviewer for their comment and have added Section 3.1 “Failure Cases and Challenges” to the updated supplementary material.
> We show additional experiments with rolling shutter data, inaccurate camera calibration, and highly rotational motion during initialization. Additionally, we describe potential failures due to dynamic objects and the challenge of neural network generalization.

---

### Official Review · Reviewer_mvF2 · 2021-07-24

**Originality:** Good
**Technical Quality:** Very Good
**Clarity Of Presentation:** Good
**Impact:** 3

**Recommendation:**

Weak Accept: I recommend accepting the paper, but will not argue for my recommendation if the majority of other reviewers have a different opinion.

**Summary:**

The paper addresses the problem of visual odometry, dense depth estimation, and volumetric mapping with a monocular camera. The proposed method uses a direct sparse windowed optimization back-end and dense tracking front-end for visual odometry; CVA-MVSNet based on multi-view stereo and a neural network architecture for dense depth estimation; and a truncated signed distance function to represent the 3D map. Experiments are performed to evaluate both the quality of visual odometry and 3D reconstruction on public datasets.

**Issues:**

Other specific comments:
- "... over RGB-D [1] or LiDAR-based [4] solutions: They are not limited to indoor environments, they are not limited to near-range sensing and they..." This seems to imply that, e.g., LiDAR is limited to near-range sensing, when LiDAR can get to up to in the order of 100m, when the camera is much more limited. Please rephrase.
- The related work section could tie more with the proposed work; for example, in 3D reconstruction, the link is explicit; the link is not explicit for monocular SLAM.
- to have a more complete evaluation of the visual odometry, feature-based SLAM system, as ORB-SLAM could be included in the comparison.
- justification of the results are important to provide, e.g., why CVA-MVSNet achieves better results with the model trained on the synthetic Replica?

Minor typos/presentation problems:
- page 2, line 58, "to weight" -> "to weigh"
- page 6, line 213 "TANDEM is a dense SLAM system that estimates camera poses in real-time.", this is a repetition -- at this point TANDEM has been already introduced multiple times, thus the sentence can be deleted.
- Table 5 appears better if not at the end of the page.

**Reviewer Expertise:**

Very good: Comprehensive knowledge of the area

**Strengths And Weaknesses:**

The paper proposes an interesting integration of a traditional visual SLAM pipeline with a learning-based approach for 3D reconstruction, displaying positive results compared to other methods. The experiments are quite complete as different parts of the pipeline are evaluated and an ablation study is included.

One of the main comments regards the fact that a dense SLAM system is proposed, that eventually could be deployed on a robot, however currently it has been tested on a powerful workstation, limiting the potential applicability on mobile robots. The paper could discuss about deployments on embedded systems.

**Summary Of Recommendation:**

Overall, while the paper presents a number of comments, including that it could be more explicit on the possibility to deploy the system on embedded systems, the paper presents a technically sound integration that shows an improvement with respect to other methods in visual SLAM and 3D reconstruction with a monocular camera.

---

> ### Author Response · Authors · 2021-08-26
> **Response to Reviewer mvF2**
>
> We thank the reviewer for their helpful and insightful comments and address the mentioned issues below. We have added significant new material to the supplementary material and will update the main paper if the paper is accepted such that we can aggregate all suggestions and change the layout of the main paper only once.
>
>
> **Computational Cost and Deployment on an Embedded Device**
>
> We thank the reviewer for their comment and acknowledge the importance of the topic. We have added Section 5.3 “Deployment on an Embedded Device” in the supplementary material, which we summarize here for convenience.
> An embedded system can benefit from software optimizations such as 16-bit float inference or NVIDIA TensorRT, which can bring speedups of up to 2x and 2.5x, respectively. However, we consider these engineering-focused optimizations outside the scope of this work.
> We discuss possible runtime performance optimizations in the supplementary material, which include: Caching the 2D feature maps of past keyframes and inference on a lower resolution, 256 x 192 instead of 640 x 480. The combination of all optimizations reduces the inference time from 158ms to 25.6ms, which makes TANDEM more feasible on embedded hardware like the Nvidia Jetson AGX Xavier without using TensorRT. Additionally, the low-resolution model does not require retraining due to the fully-convolutional and geometry-based CVA-MVSNet and suffers only a small drop in depth estimation accuracy from 94.40% to 91.43% on EuRoC.
>
>
> Further VO comparisons:
> We run ORB-SLAM on the EuRoC sequences and disable loop closure and relocalization, i.e. the full-slam mode, for a fair comparison. It achieves an average ATE RMSE of 0.67 in comparison to 0.12 for TANDEM and 0.17 for DSO. Additionally, we have added a comparison with DeepTAM in the supplementary material, which shows that TANDEM produces more accurate poses, although DeepTAM requires ground truth depth maps. Please also consider the answer to reviewer W394 for further details.
>
> **Dataset Comparison**
>
> The ScanNet dataset shows lower photometric quality than the replica dataset and forces the network to learn smoother depth maps to account for the low data quality. Specifically, the pseudo ground truth poses estimated by BundleFusion and the intrinsic parameters are not photometrically consistent (cf. Azinovic et al., “Neural RGB-D Surface Reconstruction” Figure 5). Additionally, the pseudo ground truth depth maps estimated by BundleFusion as well as the depth maps from the RGB-D sensor are not complete and not perfectly accurate (cf. Azinovic et al., “Neural RGB-D Surface Reconstruction” Figure 4). Finally, the images show considerable white-balancing issues, different exposure times, motion blur, and rolling shutter effects.
>
>
> **Comments Regarding Wording, Related Work, and Typos**
>
> We thank the reviewer very much for their extensive comments regarding the manuscript. We will incorporate the proposed changes if the paper is accepted such that we can aggregate all suggestions and change the layout of the main paper only once.

---

> > ### Comment · Reviewer_mvF2 · 2021-08-31
> > **Thanks for the reponse**
> >
> > Thanks for the provided response that clarifies the points raised in the review. It is good that at least a discussion is provided in terms of inference time and how to reduce it so that the proposed method can run on an embedded system. It would be great to have an actual experiment with an embedded system, given they are low-cost and given that the paper focuses on real-time dense 3D mapping. Overall, this review remains on the positive side.

---

### Official Review · Reviewer_XeDS · 2021-07-26

**Originality:** Good
**Technical Quality:** Good
**Clarity Of Presentation:** Good
**Impact:** 3

**Recommendation:**

Weak Accept: I recommend accepting the paper, but will not argue for my recommendation if the majority of other reviewers have a different opinion.

**Summary:**

The paper presents an approach for dense tracking and mapping. Based on DSO, it performs photometric bundle adjustment to estimate the pose of keyframes in a sliding window. Poses and images are then forward to a cost-volume based learned module that returns the depth for every keyframe. The depth are then used to create a global TSDF-based representation of the environment. From this, depths a rendered to improve camera tracking in the front end.

**Issues:**

* The improved performance due to the rendered depth maps could be investigated more.
* The order of tables and figures could better match the text

**Reviewer Expertise:**

Good: General knowledge of the area

**Strengths And Weaknesses:**

The overall systems seems well engineered. It draws on state-of-the-art methods and the presented results are competitive.

From a SLAM perspective, there is only a minor change when comparing to DSO. This is the use of the rendered depth (from the TSDF) for the front-end tracking. However, looking at the results, this adaptation seems to have a relevant effect on the overall system. This may be due to the increased image region that can be tracked against. It would be good to have a discussion on this somewhere.

In terms of reconstruction, the contribution is fairly incremental and mainly lies in combining prior work. This lies in adapting Cas-MVSNet to a visual aggregation module, and to merge all estimated depths into a TSDF.

One interesting aspect is the applicability of the paper to other data than the one it was trained on. On the SLAM side this could just mean that the tracking would fall back to DSO only, if the predicted depth become to inaccurate (some experiments to check this would be great).

The structure of the paper is good and the literature review appears complete.

**Summary Of Recommendation:**

While the contribution is fairly incremental, the overall system seems to be performing well and both the increased performance of the front-end and the applicability of the approach to data it was not trained on is interesting.

---

> ### Author Response · Authors · 2021-08-26
> **Response to Reviewer XeDS**
>
> We thank the reviewer for their helpful comments and would like to point towards the substantially expanded supplementary material. We will change the order of tables and figures in the main paper if the paper is accepted such that we can aggregate all suggestions and change the layout of the main paper only once.
>
>
> **Improved Performance Due to Rendered Depth Maps**
>
> We thank the reviewer for their comment and have added Figure 5 and Figure 6 in the supplementary material that show the depth maps used for dense and sparse tracking. We would also like to point toward Strasdat et al. “Visual SLAM: Why filter?” ([16] in the new supplementary) and Newcombe “Dense visual SLAM” ([17] in the new supplementary), which show that using dense tracking can increase robustness and accuracy.
>
>
> **Generalization**
>
> We agree with the reviewer that the applicability to non-training datasets is crucial. Recent dense SLAM systems leveraging deep neural networks mostly rely on monocular depth estimation, e.g., CNN-SLAM, CodeSLAM, DeepFactors, etc. However, monocular depth estimation is known to have limited accuracy and, more seriously, weak generalization capability which is crucial to deploy SLAM systems in practice. TANDEM, for the first time, tries to tackle the dense SLAM problem with deep multi-view stereo (MVS) instead of monocular depth estimation. We believe combining classical VO with deep MVS takes the advantage of both worlds -- pose estimation is more reliable using classical optimization while dense depth estimation shows higher quality, especially in terms of completeness, using deep neural networks when the poses are given. However, further improving the generalization capability is a fruitful direction for future work, which we mention in Section 3.1 of the supplementary material.

---

> > ### Comment · Reviewer_XeDS · 2021-09-03
> > **Response to Authors**
> >
> > The authors have addressed most of my concerns and I will keep my rating as is.

---

### Meta-Review · Area_Chair_kNsW · 2021-08-11

**Recommendation:** Accept (Poster)
**Confidence:** 5

**Metareview:**

The paper studies an interesting problem and provides a well-engineered solution. The reviewers appreciated the technical approach, the clarity of the presentation, the experimental evaluation, and the applicability of the approach outside the training regime.
The main weaknesses are:
- The approach is incremental and relies on a small modification of DSO and Cas-MVSNet. However, the resulting system provides a compelling integration of classical VO and deep MVS, which can trigger further work from the community.
- The computational cost of the approach might prevent applications on an actual robot / embedded system. After the rebuttal, the authors added further discussion about potential runtime optimization in the supplementary material, which the reviewers and area chair found satisfactory.
- The presentation in the original submission lacked details about the initialization of the TSDF, a discussion about runtime, choice of training/testing datasets, and a comparison against more recent VO/SLAM systems. These shortcomings have been resolved in the revised paper.

---

> ### Author Response · Authors · 2021-08-26
> **Response to Meta Review**
>
> We thank the meta reviewer and all reviewers for their valuable comments, which we will address in the following and in the individual responses. We have added new experiments and clarifications to the updated supplementary material and will reference them from within the responses. The comments regarding the main paper are highly appreciated and we will incorporate them if the paper is accepted such that we can aggregate all suggestions and change the main paper only once.
>
>
> **Novelty**
>
> We appreciate the reviewers pointing out the limitations on our technical novelty. We would like to emphasize that the significant contribution of TANDEM does not lie in the incremental improvement over the prior works (DSO and Cas-MVSNet), but rather in the novel system design of deep learning boosted dense SLAM which is practical, generalizable, and accurate. Recent dense SLAM systems leveraging deep neural networks mostly rely on monocular depth estimation, e.g., CNN-SLAM, CodeSLAM, DeepFactors, etc. However, monocular depth estimation is known to have limited accuracy and, more seriously, weak generalization capability which is crucial to deploy SLAM systems in practice. TANDEM, for the first time, tries to tackle the dense SLAM problem with deep multi-view stereo (MVS) instead of monocular depth estimation. We believe combining classical VO with deep MVS takes the advantage of both worlds -- pose estimation is more reliable using classical optimization while dense depth estimation shows higher quality, especially in terms of completeness, using deep neural networks when the poses are given. Our extensive experiments have shown that such a novel design leads to superior performance to the prior works. In fact, the incremental improvements on DSO and Cas-MVSNet come naturally when we try to complete the design and bridge the gap between DSO and Cas-MVSNet by making the two components take advantage of each other. As a result, by using dense front-end tracking with the depth from MVS, TANDEM shows higher robustness on camera tracking while retaining real-time performance. And, by effectively using the whole keyframe window from visual odometry with adaptive view aggregation, the proposed CVA-MVSNet shows higher reconstruction quality than prior works. We believe that the design choices of TANDEM contribute notable novelties and would also inspire the community for the future development of dense SLAM.
>
>
> **Computational Cost and Deployment on an Embedded Device**
>
> Please consider the newly added Section 5.3 in the supplementary material, which we summarize here for convenience. In this section we discuss possible runtime performance optimizations, which include: Caching the 2D feature maps of past keyframes and inference on a lower resolution, 256 x 192 instead of 640 x 480. The combination of all optimizations reduces the inference time from 158ms to 25.6ms, which makes TANDEM more feasible on embedded hardware like the Nvidia Jetson AGX Xavier without using TensorRT. Additionally, the low-resolution model does not require retraining due to the fully-convolutional and geometry-based CVA-MVSNet and suffers only a small drop in depth estimation accuracy from 94.40% to 91.43% on EuRoC.
>
>
> **TSDF Initialization**
>
> We have added a discussion of the TSDF initialization in Sec. 2 of the supplementary material. The TSDF volume is initialized to represent empty space, i.e. with each voxel having a weight of zero. Because we use voxel hashing [29], this means that initially no voxel blocks are allocated. As our paper states in line 119 ff. the initial dense tracking will thus rely purely on sparse points from the photometric bundle adjustment and dense depth will be utilized after the first depth map is integrated into the TSDF volume. We visualize this in Figure 5 in the supp..
>
> **Training & Testing Datasets**
>
> The training and testing datasets are discussed in Sec. 4.1 “Datasets” of the main paper and we have already evaluated a model trained only on the ScanNet dataset to provide a fair comparison with other methods only trained on ScanNet, e.g. DeepFactors and CodeVIO. Specifically, the columns “Ours (ScanNet)” in Table 2 and Table 3 show the results of our model trained on ScanNet only.  We will clarify this in the text of the main paper.
>
>
> **Comparison Against More Recent SLAM/VO Methods**
>
> The main paper compares TANDEM with CodeVIO, which was published in ICRA 2021. We have added the requested comparison with DeepTAM in Tab. 1 of the supp.. Please refer to the response to reviewer W394 for more details about the DeepTAM comparison. We show comparisons with ORB-SLAM and D3VO in the responses to reviewer mvF2 and W394, respectively.
>
>
> **Comparative Runtime Analysis**
>
> We directly compare the runtimes of TANDEM, DeepFactors, and DSO on the same hardware in the response to reviewer xCmq. All three methods achieve real-time performance on EuRoC. TANDEM and DeepFactors show similar runtimes while DSO runs much faster.

---

### Decision · Program_Chairs · 2021-09-13

**Decision:**

Accept (Poster)

**Comment:**

The paper studies an interesting problem and provides a well-engineered solution. The reviewers appreciated the technical approach, the clarity of the presentation, the experimental evaluation, and the applicability of the approach outside the training regime.
The main weaknesses are:
- The approach is incremental and relies on a small modification of DSO and Cas-MVSNet. However, the resulting system provides a compelling integration of classical VO and deep MVS, which can trigger further work from the community.
- The computational cost of the approach might prevent applications on an actual robot / embedded system. After the rebuttal, the authors added further discussion about potential runtime optimization in the supplementary material, which the reviewers and area chair found satisfactory.
- The presentation in the original submission lacked details about the initialization of the TSDF, a discussion about runtime, choice of training/testing datasets, and a comparison against more recent VO/SLAM systems. These shortcomings have been resolved in the revised paper.